# Can transcranial direct current stimulation (tDCS) over the motor cortex increase endurance running performance? a randomized crossover-controlled trial

Géraldine Martens[1,2,3]*, Stéphanie Hody[1], Stephen Bornheim[1], Luca Angius[4], Louis De Beaumont[2,3], Felipe Fregni[5], Giulio Ruffini[6], Jean-François Kaux[1,7], Aurore Thibaut[8], Thierry Bury[1]

1 Department of Physical Activity and Rehabilitation Sciences, University of Liege, Liege, Belgium, 2 Sport & Trauma Applied Research Lab, Montreal Sacred Heart Hospital Research Center, Montreal, Quebec, Canada, 3 Department of Surgery, University of Montreal, Montreal, Quebec, Canada, 4 Faculty of Health and Life Sciences, Department of Sport, Exercise and Rehabilitation, Northumbria University, Newcastle upon Tyne, United Kingdom, 5 Neuromodulation Center and Center for Clinical Research Learning, Spaulding Rehabilitation, Hospital and Massachusetts General Hospital, Harvard Medical School, Boston, Massachusetts, United States of America, 6 Neuroelectrics Barcelona, Barcelona, Spain, 7 Department of Physical Medicine and Sports Traumatology, SportS2, FIFA Medical Centre of Excellence, IOC Research Centre for Prevention of Injury and Protection of Athlete Health, FIMS Collaborative Center of Sports Medicine, University and University Hospital of Liège, Liège, Belgium, 8 Coma Science Group, GIGA Consciousness, University and University Hospital of Liege, Liege, Belgium

* geraldine.martens@uliege.be

## Abstract

Transcranial direct current stimulation (tDCS), a non-invasive brain stimulation technique, has been shown to increase exercise performance in strength and cycling studies but its effects on running endurance remain unclear. The objectives of this randomized sham-controlled crossover trial were to assess tDCS efficacy on submaximal treadmill running time to exhaustion (TTE). Forty-five healthy male runners aged between 18 and 32 years (mean maximal oxygen consumption: 46.6 mL/min/kg) performed two constant-load tests at 90% of their maximal aerobic speed preceded by 20 minutes of active or sham multi-channel (5 anodes, 3 cathodes) tDCS applied over the bilateral motor cortex with a total intensity of 4 mA. Ratings of perceived exertion (RPE), blood lactate, $VO_2$, and heart rate were monitored every five minutes until volitional exhaustion. The median [IQR] TTE was similar following active (12.2 [10.5, 16.1] minutes) or sham (12.5 [10.2, 15.1] minutes) tDCS (p = 0.96). Likewise, there were no significant differences between active and sham conditions for RPE, blood lactate, final $VO_2$, and final heart rate (all p $\geq$ 0.05). No difference in TTE was found when stratifying groups according to their $VO_2max$ (i.e., $VO_2max \geq$ 45 mL/min/Kg, p = 0.53; $VO_2max <$ 45 mL/min/Kg, p = 0.45) but there was a trend for a significant correlation between $VO_2max$ and change in TTE (p = 0.06). TDCS applied over the bilateral motor cortex did not improve endurance performance in a large sample of trained runners. Characterization of individual tDCS responsiveness deserves further consideration. In our experimental conditions, tDCS had no ergogenic effect on endurance running performance.

**Data Availability Statement:** All relevant data are within the manuscript and its Supporting Information files.

**Funding:** The author(s) received no specific funding for this work.

**Competing interests:** The authors have declared that no competing interests exist.

**Trial registration**: Clinical trial registration: NCT04005846.

## 1. Introduction

Increasing exercise endurance through external interventions is raising interest. Neuromodulation techniques to increase brain activity and, possibly, exercise performance have been tested; however, their efficacy and ethical use have been questioned [1–3]. Among them, transcranial direct current stimulation (tDCS) stands out given its affordability, safety and ease of use. TDCS uses low intensity electrical current applied on the scalp to modulate the neuronal excitability of the targeted brain areas [4]. The effects depend on the current's flow direction: anodal stimulation tends to increase the excitability in standard protocols (20 minutes– 2 mA) while cathodal stimulation tends to decrease it [5]. These effects can be direct, through the online electrical alteration of the resting membrane potential, or indirect, through offline long-term potentiation/depression-like mechanisms[6]. Regarding endurance performance, tDCS applied over the primary motor cortex (M1) could potentially increase its cortical excitability, facilitate the supraspinal drive, reduce central fatigue, and prolong muscular endurance [7]. These hypotheses have, however, not been confirmed since current evidence fail to establish a causal link between corticospinal-motoneuronal excitability and improvement in exercise performance [8,9].

Endurance can be quantified by time to exhaustion (TTE) trials, a marker of capacity represented by the length of time that a given power output can be maintained [10]. Another key component of endurance performance relates to the perceived exertion during exercise, one of the most important features of fatigue, typically measured with self-reported rating of perceived exertion (RPE) [11].

Several studies investigated tDCS effects on athletic performance in various settings (e.g., isometric or isokinetic strength, cycling, shooting) and with different populations (e.g., non-athletes, amateurs, professional athletes), leading to mixed results [12–15], with about 60% of studies reporting physical performances improvements [16]. Heterogeneity among study protocols and variability in individual response to tDCS could underlie such discrepancies across study results. Regarding stimulation intensity, most studies used 1.5 or 2 mA current intensities, which limits the evaluation of tDCS dose effects in this context, while higher intensities (e.g., 4 mA) could hold better efficacy and are considered safe [17]. Another variable that may affect the response to tDCS is the athlete's baseline athletic level. A prior strength study suggested that individuals with a lower level of endurance capacity might benefit more from the ergogenic effects of M1-tDCS [18], possibly due to a ceiling effect in trained athletes. This hypothesis still needs to be confirmed in whole-body exercises (i.e., cycling, running) [12].

Recent meta-analyses show a significant beneficial effect of tDCS on performance as measured by TTE in whole-body exercises [13–15]. Among retrieved studies, only three trials focused on the running modality, although running tests to exhaustion elicit higher levels of oxygen consumption and energetic demands relative to cycling as the former exercise involves more muscular mass [19,20]. On one hand, a single-blind randomized controlled trial investigating the effects of a single session of M1-tDCS using a commercial device in 10 trained runners showed a significant increase in TTE following active (21 minutes) compared to sham (18 minutes) stimulation in the absence of significant changes in RPE and cardiorespiratory variables [21]. On the other hand, a double-blind randomized controlled trial with an identical tDCS protocol in 13 recreational runners reported no significant difference in TTE or RPE

change between active (9 minutes) and sham (9 minutes) stimulation [22]. Another single-blind study used a commercial device and reported a higher peak oxygen consumption following active versus sham tDCS in 17 physically active men [23]. Besides being limited by their restricted sample sizes, these studies failed to provide an additional control task condition without tDCS in order to account for a potential placebo effect related to an external intervention (i.e., active or sham tDCS application).

Characterizing ergogenic effects of tDCS requires an objective monitoring of exercise-related physiological variables. Oxygen consumption and blood lactate accumulation are key markers of aerobic capacity and exercise intensity, respectively [24,25]. Oxygen consumption kinetics during constant-load endurance running are typically stable and do not differ between active and sham tDCS conditions [21,22]. The blood lactate increase is more sensitive to exercise intensity and accumulates significantly more, along with TTE, following motor and prefrontal tDCS, as compared to cathodal and sham, in cycling TTE trials on small samples (i.e., n≤12) [26,27]. Such effects remain to be confirmed with larger samples in running tDCS studies.

To address these gaps, we conducted a large randomized sham-controlled clinical trial to investigate the effects of a single session of M1-tDCS on running endurance performance, measured with TTE, in trained athletes while controlling for perceived exertion and performance-related physiological parameters. We hypothesized that performance would be increased following active and not sham tDCS, and that participants with lower baseline athletic levels will benefit more from tDCS.

## 2. Material and methods

### 2.1 Standard protocol approvals, patient consent and study registration

The study was approved by the institutional ethics committee (Comité d'Éthique Hospitalo-Facultaire Universitaire de Liège, approval number CE2019/186) before its beginning. Written informed consents were obtained for all of the participants. The study was registered as a clinical trial (ClinicialTrials.gov NCT04005846), conducted in accordance with the Declaration of Helsinki and reported following the CONSORT guidelines (S1 Table). The study protocol can be found as (S1 File). All study-related data was managed and stored in accordance with the EU General Data Protection Regulation.

### 2.2 Participants

Between October 4, 2019 and March 24, 2021, we recruited healthy males aged between 18 and 35, training in endurance sports for at least two hours a week, with a maximal oxygen consumption ($VO_2$max) comprised between 30 and 65 mL/min/kg, which was determined on the first screening visit. Exclusion criteria were: smoking, dietary supplementation, coffee consumption above 10 units a week, alcohol consumption above 4 units a week, centrally-acting medication and history of pain or lesion of the lower limbs in the past six weeks. Additionally, the tDCS Safety Screening Tool (TSST) was completed [28]. Participants were recruited among students of the faculty, in affiliated sports clubs and through the Physiology Laboratory database. The CONSORT Participant flow diagram is presented in Fig 1.

### 2.3 Sample size estimation

The study sample size was estimated a priori using Student's t-tests (S3 Table) based on previous studies with amateur and competitive participants (cyclists and runners) [22,26,29–32]. It reached 52 subjects based on an effect size of 0.8 and a power of 80% with an alpha level of 5%.

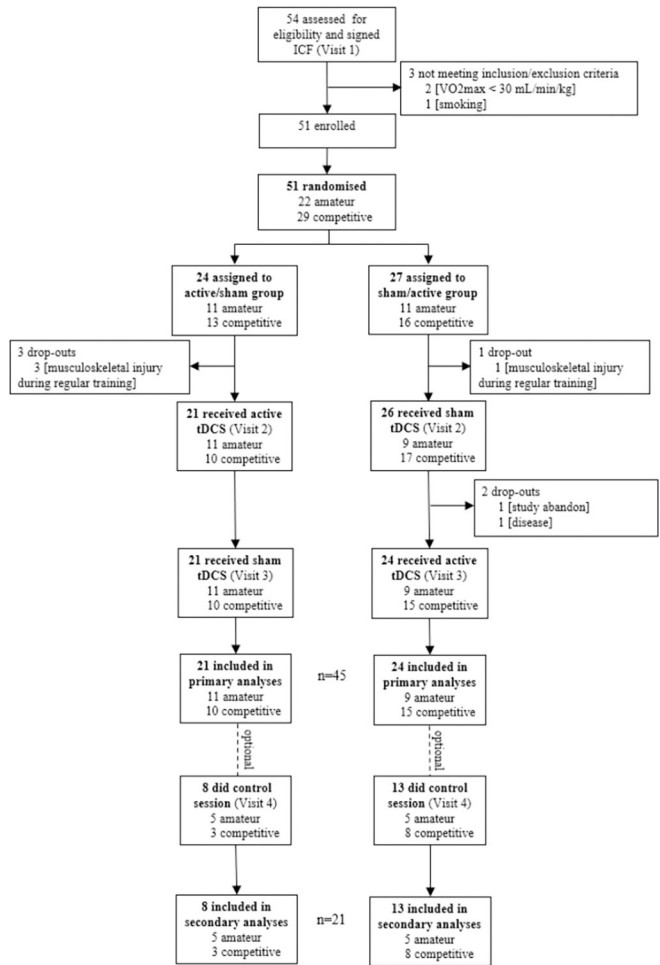

**Fig 1. Flow diagram of study participants.**

## 2.4 Procedures

This was a randomized double-blind sham-controlled crossover trial with a screening visit, two consecutive tDCS (active/sham randomized) followed by constant-load tests, and a control constant-load test (without tDCS), all spaced by seven days.

**2.4.1 Screening visit (Visit 1).** During the screening visit, participants performed a $VO_2max$ test conducted as described in Martens et al. [33] and inspired from the Bruce protocol [34]. Following the warm up (5 minutes at 8 km/h), the pace was incremented by 2 km/h every 2 minutes up to 16 km/h and then by 1 km/h every 3 minutes until participants' exhaustion. Strong verbal encouragements were provided and the maximal nature of the effort was determined using the following criteria: heart rate above 90% of the age-predicted maximum (i.e., 220-age); respiratory exchange ratio $\geq$ 1.10; plateau in oxygen uptake ($VO_2$) and; lactate blood level $\geq$ 8 mmol/L. $VO_2$, respiratory exchange ratio and heart rate (HR) were measured continuously (Ergostick, Geratherm Respiratory, Germany and Polar Belt, Polar, USA). Lactate blood levels and RPE were measured at the end of each increment using capillary blood (YSI 1500 Sport L-Lactate Analyser, YSI, USA) and the French validated version of the Borg's 6–20 scale [35], respectively. The maximal aerobic speed (MAS) was defined as the highest speed achieved at $VO_2max$.

**2.4.2 tDCS and TTE trials (Visits 2 and 3).** Following active or sham tDCS (randomized order, crossover design) and a warm up (5 minutes at 8 km/h), participants performed a constant-load TTE trial at 90% of their MAS until volitional exhaustion or inability to keep the pace. Using the above-mentioned equipment, cardiorespiratory parameters were continuously measured while lactate blood levels and RPE were collected every 5 minutes and at the end of the trial. The tDCS intervention consisted of 20 minutes of stimulation at 4 mA preceded and followed by a 30-second ramping period (active condition) or a 30-second ramping period only (sham condition) with a built-in double-blind mode (details below). A topical anaesthetic cream was applied over the stimulation area prior to each session to diminish somatosensory perception of the stimulation and to ensure blinding. During tDCS, participants were seated on a chair and instructed to remain calm and alert. The Stimweaver multichannel tDCS montage optimization algorithm was used to target the bilateral motor network [36]. Intensity was set at maximum 2 mA per anode for a maximum total injected current of 4 mA (Fig 2). Using the Starstim 8 system (Neuroelectrics, Spain) with Ag/AgCl 3.14 cm$^2$ electrodes and conduction gel, five anodes were placed over C1, C2, C3, C4, Cz and three cathodes over P3, P4, Fz (international 10–20 EEG system [37]).

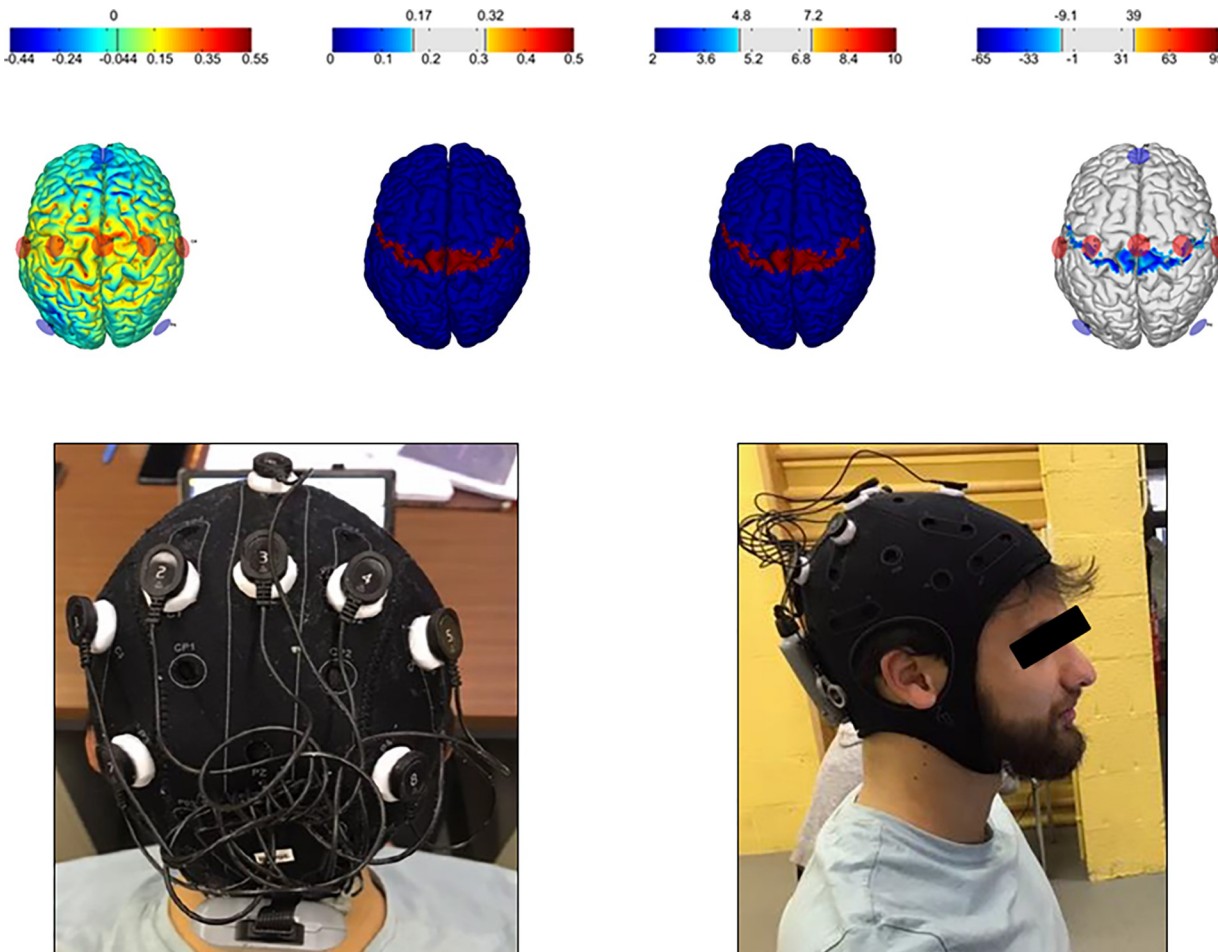

**Fig 2. tDCS montage.** E-Field (normal to cortical surface, in V/m) and current density modelling (provided by Neuroelectric©) with anodes in red and cathodes in blue (upper part); montage on the cap before stimulation (lower part).

At the end of the visit, participants filled out a questionnaire about their perception of receiving active versus sham tDCS with a 5-point certainty grading scale (1: not sure at all; 5: absolutely sure). A second questionnaire assessed potential adverse effects using an evidence-based list of potential symptoms, a self-reported severity value and a certainty grade of whether the reported effects are related to tDCS or not [38].

**2.4.3 Control session (Visit 4).** To isolate potential placebo effects related solely to the external intervention (i.e., active or sham tDCS), a fourth visit consisting only of the constant-load TTE trial at 90% MAS (without tDCS) was conducted. When co-constructing the protocol with participating athletic associations, the addition of a fourth TTE trial over a 1-month period raised feasibility issues due to training programs constraints. Consequently, the latter session was presented as optional to the participating athletes.

All laboratory visits were performed at the same moment of the week, on the same treadmill, with the same running gear, in a temperature-controlled room and with the same investigators to ensure reproducibility.

## 2.5 Randomization and blinding

Simple randomization was performed by a third party using a computer-generated sequence in a 1:1 allocation ratio. The tDCS device was used in double-blind mode and pre-programmed with either active or sham-coded sessions provided to the investigators before each test. Neither the investigators nor the participants were aware of the coded allocation and the tDCS software depicted identical information (including identical stimulation time) for both active and sham conditions. Upon study completion and database cleaning, the code was provided to the person in charge of the analyses.

## 2.6 Outcomes

The primary outcome was the TTE following active versus sham tDCS (i.e., tDCS efficacy) at the group level and in subgroups stratified according to fitness level (amateur, competitive) based on the median $VO_2max$. The secondary outcomes were: (1) the tDCS efficacy in the subgroup who completed the control session (Visit 4) and; the influence of fitness level ($VO_2max$) on tDCS efficacy.

## 2.7 Statistical analyses

Statistical analyses were performed per protocol (i.e., dropouts excluded) using R4.2.1 (R Foundation for Statistical Computing, Vienna, Austria). The normality of the data distribution was assessed using Shapiro-Wilk tests. According to the nature of the distribution, mean and standard deviation or median and interquartile ranges were used for descriptive analyses. Baseline comparisons (age, weight, height, body fat, years of training, weekly training, $VO_2max$, MAS, TTE on $VO_2max$ test, maximal heart rate, maximal respiratory exchange ratio) between groups (active/sham and sham/active) were performed using independent Student's t-tests (normal distribution) and Wilcoxon rank-sum tests (non-normal distribution). Comparisons between active and sham tDCS sessions were then performed using paired Student's t-tests and Wilcoxon signed-rank tests. Subgroup analyses including the additional control condition (Visit 4) were performed using one-way ANOVAs (normal distribution) or Kruskal-Wallis tests (non-normal distribution). A new variable derived from the time (TTE) by treatment (active-sham condition) interaction was constructed to quantify the change in TTE between active and sham tDCS: ΔTTE = TTE active minus TTE sham. The relationship between baseline $VO_2max$ and ΔTTE was assessed using Spearman's correlations. Results were considered significant at $p < 0.05$.

**Table 1. Sociodemographic and aerobic profile of the study participants.**

| | Total (N = 45) | Active First (N = 21) | Sham First (N = 24) | p-value [a] |
|---|---|---|---|---|
| **Age (years)** | | | | |
| Median [Q1, Q3] | 21.0 [21.0, 22.0] | 21.0 [21.0, 22.0] | 21.0 [20.8, 22.3] | 0.647 |
| **Weight (Kg)** | | | | |
| Mean (SD) | 73.1 (8.9) | 74.2 (8.2) | 72.2 (9.6) | 0.466 |
| **Height (cm)** | | | | |
| Mean (SD) | 179 (7.2) | 179 (8.1) | 179 (6.6) | 0.971 |
| **Body fat (%)** | | | | |
| Mean (SD) | 13.7 (3.4) | 14.0 (3.6) | 13.5 (3.4) | 0.650 |
| **Years of training** | | | | |
| Median [Q1, Q3] | 7.0 [3.0, 10.0] | 7.0 [2.0, 12.0] | 7.3 [4.6, 10.0] | 0.632 |
| **Weekly training (hours)** | | | | |
| Median [Q1, Q3] | 3.0 [2.0, 6.0] | 2.0 [2.0, 7.0] | 3.0 [2.0, 4.1] | 0.818 |
| **VO$_2$max (mL/min/Kg)** | | | | |
| Mean (SD) | 46.6 (7.5) | 45.2 (7.6) | 47.9 (7.3) | 0.228 |
| **Max. aerobic speed (Km/h)** | | | | |
| Median [Q1, Q3] | 16.0 [14.0, 16.5] | 14.4 [14.0, 16.3] | 16.0 [14.4, 16.6] | 0.199 |
| **TTE on VO$_2$max task (minutes)** | | | | |
| Mean (SD) | 11.6 (3.3) | 11.1 (3.6) | 12.1 (2.9) | 0.281 |
| **Max. heart rate (bpm)** | | | | |
| Mean (SD) | 194.8 (9.4) | 194.8 (11.2) | 194.8 (7.8) | 0.994 |
| **Max. respiratory exchange ratio** | | | | |
| Median [Q1, Q3] | 1.2 [1.2, 1.3] | 1.2 [1.2, 1.3] | 1.2 [1.2, 1.3] | 0.873 |

VO$_2$max = maximal oxygen consumption; TTE = time to exhaustion.

[a] = Student's t-test or Wilcoxon test according to data distribution.

## 3. Results

Fifty-four participants were included; 45 completed the three protocol visits, and 21 completed the additional control visit. The flowchart is presented in Fig 1 and the demographics and aerobic profiles are presented in Table 1, with no significant differences between allocation groups. Based on the median VO$_2$max (46.7), participants with a VO2max $\geq$ 45 mL/min/Kg (n = 25) were classified as competitive while participants <45 (n = 20) were classified as amateur. The study dataset is available as (S1 Dataset).

### 3.1 tDCS efficacy

At the group level (n = 45), the median [IQR] TTE was 12.2 [10.5, 16.2] minutes after active tDCS and 12.5 [10.2, 15.1] after sham tDCS (Fig 3A). There was no significant difference between active and sham conditions (V = 523; p = 0.96). The final RPE, blood lactate, VO$_2$ and heart rate (Fig 3) were also similar between active and sham conditions (all p's >0.05), as presented in Table 2. In the subgroup of amateur participants (n = 20), the TTE was 10.7 [9.4, 13.8] minutes following active tDCS and 10.3 [9.6, 12.8] following sham, with no significant difference between conditions (V = 126; p = 0.45). For the competitive subgroup (n = 25), the

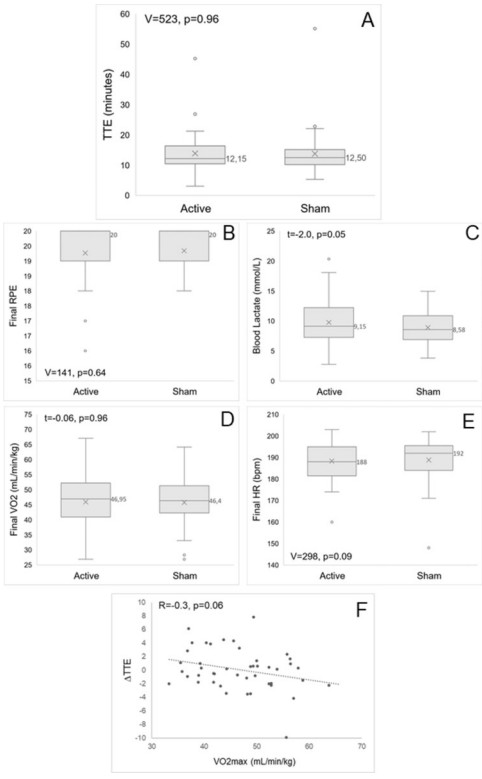

**Fig 3. Study outcomes.** Differences in A. time to exhaustion (TTE); B. final rating of perceived exertion (RPE); C. blood lactate; D. final oxygen consumption (VO2) and E. final heart rate (HR) between active and sham tDCS conditions. Influence of baseline athletic characteristics on improvement in time to exhaustion (ΔTTE = TTE active tDCS minus TTE sham): F. maximal oxygen consumption (VO2max).

TTE was 15.2 [11.7, 16.7] minutes for active tDCS and 15.0 [12.5, 15.7] for sham, with no significant difference between conditions (V = 138; p = 0.53). The RPE and physiological variables were also similar between active and sham conditions for both subgroups (all p's >0.05, see Table 2).

In the subgroup of participants who completed the 4th control session without tDCS intervention (n = 21, secondary analysis), the mean ± SD TTE was 13.3 ± 3.5 minutes after active tDCS, 13.1 ± 3.1 after sham tDCS and 12.6 ± 4.0 for the control session. There was no significant difference between the three conditions (F = 0.19; p = 0.83). The final RPE, blood lactate, $VO_2$ and heart rate were also similar between active, sham and control conditions (all p >0.05), as presented in Table 3. In the subgroup of amateur participants (n = 10), the TTE was 11.7 ± 3.3 minutes following active tDCS, 11.2 ± 2.3 minutes following sham and 10.5 ± 3.7 minutes for the control session, with no significant difference between conditions (F = 0.35; p = 0.71). For the competitive subgroup (n = 21), the TTE was 14.7 ± 3.1 minutes for active tDCS, 14.8 ± 2.6 minutes for sham and 14.5 ± 3.4 minutes for the control session, with no significant difference between conditions (F = 0.03; p = 0.97). The RPE and physiological variables were also similar between active, sham and control conditions for both groups (all p's >0.05, see Table 3).

### 3.2 Influence of fitness level

There was no significant correlation between the individual $VO_2max$ and the change in TTE between active and sham tDCS (i.e., ΔTTE); however, a trend was noted (R = -0.30; p = 0.058,

**Table 2. Time to exhaustion (TTE, primary outcome), rating of perceived exertion (RPE), final blood lactate, final oxygen uptake (VO₂) and final heart rate (HR) following active and sham tDCS protocols in the study sample (n = 45) and in subgroups stratified according to VO₂max (i.e., amateur < 45 mL/min/Kg; n = 20 and competitive ≥ 45 mL/min/Kg; n = 25).**

| Group | Variable | Active tDCS | Sham tDCS | Statistic [a] |
|---|---|---|---|---|
| **Study sample (n = 45)** | TTE (min) | 12.2 [10.5, 16.1] | 12.5 [10.2, 15.1] | V = 523; p = 0.96 |
| | Final RPE | 20 [19, 20] | 20 [19, 20] | V = 141; p = 0.64 |
| | Lactate (mmol/L) | 9.8 ± 3.7 | 8.9 ± 2.8 | t = -2.0; p = 0.052 |
| | Final VO₂ (mL/min/kg) | 45.9 ± 8.7 | 45.8 ± 7.8 | t = -0.06; p = 0.96 |
| | Final HR (bpm) | 188 [183, 195] | 192 [184, 195] | V = 298; p = 0.09 |
| **Amateur (n = 20)** | TTE (min) | 10.7 [9.4, 13.8] | 10.3 [9.6, 12.8] | V = 126; p = 0.45 |
| | Final RPE | 20 [18, 20] | 20 [20, 20] | V = 17.5; p = 0.33 |
| | Lactate (mmol/L) | 8.3 [7.2, 10.7] | 7.8 [6.7, 9.9] | V = 124; p = 0.50 |
| | Final VO₂ (mL/min/kg) | 39.1 ± 7.4 | 40.2 ± 6.3 | t = -1.3; p = 0.21 |
| | Final HR (bpm) | 191 ± 9 | 192 ± 8 | t = -1.1; p = 0.28 |
| **Competitive (n = 25)** | TTE (min) | 15.2 [11.7, 16.7] | 15.0 [12.5, 15.7] | V = 138; p = 0.53 |
| | Final RPE | 19 [19, 20] | 19 [19, 20] | V = 47; p = 0.55 |
| | Lactate (mmol/L) | 10.0 ± 3.4 | 9.4 ± 2.3 | t = 1.7; p = 0.11 |
| | Final VO₂ (mL/min/kg) | 51.1 ± 5.3 | 50.2 ± 5.7 | t = 1.1; p = 0.28 |
| | Final HR (bpm) | 187 [183, 192] | 189 [184, 193] | V = 120; p = 0.26 |

Descriptive statistics are presented as median [IQR] or mean ± SD depending on data distribution.

[a] = Student's t-test (t) or Wilcoxon test (V) according to data distribution.

Fig 3F). The negative correlation shows that participants with less aerobic capacity (i.e., lower VO₂max) presented a larger improvement in TTE following active stimulation (i.e., greater ΔTTE).

**Table 3. Time to exhaustion (TTE), rating of perceived exertion (RPE), final blood lactate, final oxygen uptake (VO₂) and final heart rate (HR) following active and sham tDCS protocols in the subgroup who completed the additional control visit (n = 21) and in subgroups stratified according to VO₂max (i.e., amateur < 45 mL/min/Kg; n = 10 and competitive ≥ 45 mL/min/Kg; n = 11).**

| Group | Variable | Active tDCS | Sham tDCS | Control | Statistic [a] |
|---|---|---|---|---|---|
| **Subgroup with control session (n = 21)** | TTE (min) | 13.3 ± 3.5 | 13.1 ± 3.1 | 12.6 ± 4.0 | F = 0.19; p = 0.83 |
| | Final RPE | 19 [18, 20] | 20 [19, 20] | 20 [19, 20] | H = 4.39; p = 0.11 |
| | Lactate (mmol/L) | 10.1 [7.3, 12.4] | 9.1 [7.4, 10.8] | 10.0 [8.0, 12.4] | H = 1.37; p = 0.50 |
| | Final VO₂ (mL/min/kg) | 46.5 ± 7.6 | 46.4 ± 6.3 | 47.7 ± 4.8 | F = 0.19; p = 0.82 |
| | Final HR (bpm) | 191 ± 6 | 193 ± 6 | 190 ± 7 | F = 1.24; p = 0.30 |
| **Amateur (n = 10)** | TTE (min) | 11.7 ± 3.3 | 11.2 ± 2.3 | 10.5 ± 3.7 | F = 0.35; p = 0.71 |
| | Final RPE | 19 [18, 20] | 20 [20, 20] | 20 [19, 20] | H = 4.31; p = 0.12 |
| | Lactate (mmol/L) | 8.7 [7.2, 12.3] | 7.6 [6.8, 9.7] | 8.2 [7.6, 11.8] | H = 0.79; p = 0.68 |
| | Final VO₂ (mL/min/kg) | 40.8 ± 6.4 | 42.5 ± 4.8 | 44.4 ± 4.1 | F = 0.97; p = 0.40 |
| | Final HR (bpm) | 193 ± 7 | 195 ± 5 | 191 ± 6 | F = 0.75; p = 0.48 |
| **Competitive (n = 11)** | TTE (min) | 14.7 ± 3.1 | 14.8 ± 2.6 | 14.5 ± 3.4 | F = 0.03; p = 0.97 |
| | Final RPE | 19 [19, 20] | 19 [19, 20] | 20 [19, 20] | H = 1.39; p = 0.50 |
| | Lactate (mmol/L) | 10.5 ± 2.8 | 9.9 ± 2.4 | 11.0 ± 3.4 | F = 0.41; p = 0.67 |
| | Final VO₂ (mL/min/kg) | 51.7 ± 4.2 | 50.0 ± 5.4 | 50.2 ± 3.8 | F = 0.47; p = 0.63 |
| | Final HR (bpm) | 190 ± 6 | 192 ± 6 | 190 ± 7 | F = 0.47; p = 0.63 |

Descriptive statistics are presented as median [IQR] or mean ± SD depending on data distribution.

[a] = One-way ANOVA (F) or Kruskal-Wallis test (H) according to data distribution.

### 3.3 Blinding efficacy

Eight (18%) of the participants detected the active condition with a degree of certainty of 4 or 5, and 10 (22%) of the participants detected the sham condition with a degree of certainty of 4 or 5. There was no significant difference between the proportions of participants who correctly detected the active and sham tDCS (Chi-squared = 1.20, p = 0.27).

### 3.4 Adverse effects

A total of 99 adverse effects were reported: 49 after the active tDCS session, reported by 32 (71%) participants and 50 after the sham session, reported by 31 (69%) participants (S2 Table). All adverse effects were classified as mild as they did not require further action or medical intervention and did not cause distress to the participants.

## 4. Discussion

### 4.1 tDCS efficacy

This study aimed at investigating the effects of a single application of 20-minute 4 mA tDCS over the bilateral motor cortex on running performance as measured by time to exhaustion (TTE) duration using a randomized sham-controlled crossover design. Our results show that tDCS did not affect endurance running between active and sham sessions whether among amateur or competitive participants. The absence of any ergogenic effect of tDCS sharply contrasts with previous reports on cycling [26,32] and running endurance performance [21]. This could be partly explained by their small sample sizes (n<20) and/or methodological differences in tDCS application or blinding. The present study included a larger sample that was based on *a priori* estimation and used a robust trial methodology with adequate blinding. Methodological differences among tDCS studies on performance represent a common issue that has been raised by several systematic reviews [12,15,39]. Replication studies therefore appear warranted.

Previous studies using transcranial magnetic stimulation applied over M1 allowed to measure excitability changes via induction of isolated muscle contractions. Some of these demonstrated a significant increase in M1 cortical excitability related to tDCS [40,41] while others did not find any effect of tDCS on cortical excitability [42,43]. For whole-body exercise endurance, only a single study using a cycling task to failure showed significant increases in corticospinal excitability of the knee extensors following anodal stimulation as compared to sham or cathodal stimulation [26]. Our primary outcome measure was focused on performance and we did not control for tDCS-related motor system changes using motor evoked potentials notably due to feasibility constraints. This limits our understanding of tDCS mechanisms in a running setting.

Controlling for cardiorespiratory and metabolic parameters allows characterizing exercise intensity and linking it with endurance performance. Given the similarity in TTEs in all conditions, we expected observing no significant changes in the performance-related physiological parameters. This was the case for the oxygen uptake and the final heart rate, which is in line with previous running studies [21,22]. However, blood lactate levels tended to be higher in participants assigned to the active tDCS condition relative to sham participants. This finding was partially in line with previous reports from Angius et al. also showing significantly greater lactate accumulation following active tDCS but paralleled with increased performance in cycling TTE [26,27]. Future studies are therefore needed to clarify how modified circulating lactate levels could potentially influence exercise performance via tDCS-supported plasticity molecules including BDNF [44,45].

Regarding psychological factors, it is well known that perceived exertion during exercise plays a key role in endurance capacity and fatigue [11]. It has been suggested that M1-tDCS could decrease the perception of effort by modulating corollary discharges upstream of the motor cortex (e.g., supplementary motor area) in weight-lifting and cycling tasks (i.e., cycling) [26,42,46]. These mechanisms rely on the predominant processing of effort perception within the supplementary motor area [47]. Our study results show, however, no significant difference in the ratings of perceived exertion between active and sham sessions. This confirms the more recent M1-tDCS studies performed in running [21,22]. Other brain targets could reveal to be relevant in tackling motivational aspects including the insular cortex [31] and the left dorsolateral prefrontal cortex [27,48,49].

The present study also sought to control for a potential placebo effect of tDCS on performance by adding an optional 4th control session during which no tDCS was applied. Our results show that in our limited subsample of 21 participants who completed the latter control session, the TTE was comparable across all sessions, therefore invalidating any placebo effect related to tDCS application.

## 4.2 Fitness level

We accounted for the potential influence of fitness athletic level on performance changes following tDCS. Previous studies suggested that lower fine motor skills or maximal strength abilities were associated with greater improvements after tDCS as opposed to higher levels [12,50–52] but confirmation for gross motor skills was pertinent. When using maximal oxygen consumption ($VO_2$max) as a marker for fitness level, we found a trend for improvement in TTE following active tDCS (ΔTTE) close to statistical significance (p = 0.058). When using the TTE on the screening test as a marker, we found a significant negative correlation with ΔTTE, suggesting less trained runners (those with lower performance on their screening test) could benefit more from tDCS (increase their TTE) as opposed to confirmed runners (lasting longer on their screening test). This aligns with the literature on specialized motor learning, where a ceiling effect in well-trained participants prevents additional benefits from neuromodulation interventions [50,53]. Similarly, the group of competitive runners might have reached their maximal level of synaptic reorganization and would not benefit from the potential tDCS-induced M1 excitability increase while the amateur runners would still have room for improvement. However, this hypothesis did not translate into our subgroup-based analysis. There was indeed no significant tDCS effect in the two subgroups of amateur and competitive runners. Further exploring this hypothesis would require a better distinction between beginners (e.g., untrained) and professional (e.g., elite-level) runners.

## 4.3 tDCS application

Regarding the safety aspects, our study confirmed previous reports on minor adverse effects (e.g., tingling, itching) incidence, including those reported after sham stimulation. When used according to established safety criteria [54], including the careful screening of study participants [28], tDCS is a safe neuromodulation technique. Regarding blinding, its efficacy with standard sham protocols has been debated.[55] Our results show that the blinding was efficiently achieved. The use of a topical anaesthetic cream likely played a role and its use would be recommended, particularly for such high-density montages. Overall, the utilization of a multichannel montage, administering a cumulative current of 4mA, which is above current standard protocols, demonstrates apparent safety, without compromising the integrity of participant blinding.

#### 4.4 Limitations

Our main limitations pertain to the tDCS protocol. We used a single session delivered before the TTE trial for feasibility constraints, while the concurrent application of tDCS with the targeted activity may have been more efficient [18,56]. This method also prevents the investigation of the cumulative effects of tDCS using several consecutive sessions. Furthermore, the cephalic tDCS montage used might have induced effects under the cathodes, potentially interfering with the anodal stimulation. Extracephalic tDCS montages, even though challenging in terms of current density simulation, appear more efficient than cephalic ones for increasing endurance performance in cycling and therefore deserve further investigation in running trials [26].

#### 4.5 Conclusion

To conclude, no beneficial effect of M1-tDCS on running performance has been identified. The potential effect of multiple sessions remains unknown and warrants further research. The fitness level might influence tDCS response and deserves further investigation. A single application of tDCS does not appear as a relevant ergogenic aid and does not currently represent a doping threat for running.

## Supporting information

**S1 Table. CONSORT checklist.**
(PDF)

**S2 Table. Adverse effects.** Adverse effects following active and sham transcranial direct current stimulation (tDCS). All adverse effects were considered as mild (i.e., did not require medical action).
(PDF)

**S3 Table.**
(DOCX)

**S1 File. Study protocol.**
(PDF)

**S2 File. Sample size estimation.** Number of participants per group according to power and effect size.
(PDF)

**S1 Dataset.**
(XLSX)

## Acknowledgments

The authors would like to thank Ms Nadia Dardenne for the statistical guidance over the sample size estimation and the students involved in data acquisition: Anaïs Vervier, Lara Yans, Solène Denis and Florence Clermont.

## Author Contributions

**Conceptualization:** Géraldine Martens, Luca Angius, Giulio Ruffini, Jean-François Kaux, Aurore Thibaut, Thierry Bury.

**Data curation:** Géraldine Martens, Stéphanie Hody.

**Formal analysis:** Géraldine Martens, Louis De Beaumont, Felipe Fregni, Aurore Thibaut, Thierry Bury.

**Investigation:** Géraldine Martens, Stéphanie Hody.

**Methodology:** Géraldine Martens, Luca Angius, Thierry Bury.

**Project administration:** Géraldine Martens, Thierry Bury.

**Resources:** Stephen Bornheim, Giulio Ruffini, Thierry Bury.

**Supervision:** Géraldine Martens, Stéphanie Hody, Stephen Bornheim, Thierry Bury.

**Visualization:** Géraldine Martens.

**Writing – original draft:** Géraldine Martens.

**Writing – review & editing:** Stéphanie Hody, Stephen Bornheim, Luca Angius, Louis De Beaumont, Felipe Fregni, Giulio Ruffini, Jean-François Kaux, Aurore Thibaut, Thierry Bury.

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
