## [Decision Letter · Decision Letter 0]

24 Jul 2024

PONE-D-24-16260Can transcranial direct current stimulation (tDCS) over the motor cortex increase endurance running performance? A randomized crossover-controlled trialPLOS ONE

Dear Dr. Martens,

Thank you for submitting your manuscript to PLOS ONE. After careful consideration, we feel that it has merit but does not fully meet PLOS ONE’s publication criteria as it currently stands. Therefore, we invite you to submit a revised version of the manuscript that addresses the points raised during the review process.

We look forward to receiving your revised manuscript.

Kind regards,

Abdolvahed Narmashiri

Academic Editor

PLOS ONE

Journal Requirements:

2. In the online submission form, you indicated that [The dataset will be made available by the corresponding author (G. Martens, geraldine.martens@uliege.be) upon motivated request.]. 

4. We note that the original protocol that you have uploaded as a Supporting Information file contains an institutional logo. As this logo is likely copyrighted, we ask that you please remove it from this file and upload an updated version upon resubmission.

Reviewers' comments:

Reviewer's Responses to Questions

**Comments to the Author**

1. Is the manuscript technically sound, and do the data support the conclusions?

Reviewer #1: Yes

Reviewer #2: No

2. Has the statistical analysis been performed appropriately and rigorously? 

Reviewer #1: Yes

Reviewer #2: Yes

3. Have the authors made all data underlying the findings in their manuscript fully available?

Reviewer #1: No

Reviewer #2: Yes

4. Is the manuscript presented in an intelligible fashion and written in standard English?

Reviewer #1: Yes

Reviewer #2: Yes

5. Review Comments to the Author

Reviewer #1: A randomized sham-controlled crossover trial was conducted which aimed to assess transcranial direct current stimulation (tDCS) efficacy on submaximal treadmill running time to exhaustion (TTE). TTE was not statistically different in the two groups.

Minor revisions:

1- Page 10, Section 2.3: State the statistical testing method which achieves 80% power.

2- Section 2.7: The term "t-tests" is missing in the following sentence. “Comparisons between active and sham tDCS sessions were performed using paired Student’s [t-tests] and Wilcoxon tests.” Also specify the Wilcoxon tests: rank sum or signed rank.

3- To assist in the review process, add line numbering to the document.

Reviewer #2: As you mentioned the most problem in your study, is single session intervention so it seems that one session in not enough to have impact in physical performance, so it is better to emphasize in the conclusion to consider this issue in further study and try to have more sessions in their studies.

6. PLOS authors have the option to publish the peer review history of their article (what does this mean?). If published, this will include your full peer review and any attached files.

Reviewer #1: No

Reviewer #2: **Yes: **Reza Rostami

---

## [Author Response · Author response to Decision Letter 0]

2 Sep 2024

We are grateful for the constructive criticism provided by the reviewers and appreciate the time commitment. We thank the editors and reviewers for the feedback and specific comments provided on our manuscript. We have addressed those as outlined below. 

We hope this improved version will meet the Journal standards and meet the reviewers’ expectations.

Respectfully,

Géraldine Martens, on behalf of all authors

Journal Requirements:

We made sure that our manuscript meets PLOS ONE’s style requirements and made the necessary amendments, including those related to file naming and authors’ affiliations.

2. In the online submission form, you indicated that [The dataset will be made available by the corresponding author (G. Martens, geraldine.martens@uliege.be) upon motivated request.]. 

We have uploaded the dataset (excel file) as supplementary information and amended our statement on data availability accordingly. 

We included captions for our Supporting Information files at the end of our manuscript and updated the in-text citations accordingly following the Journal guidelines.

4. We note that the original protocol that you have uploaded as a Supporting Information file contains an institutional logo. As this logo is likely copyrighted, we ask that you please remove it from this file and upload an updated version upon resubmission.

We removed the logo from the file and uploaded an updated version.

We reviewed our reference list to ensure it is complete and correct and have not made any changes.

Reviewers' comments:

5. Review Comments to the Author

Reviewer #1: A randomized sham-controlled crossover trial was conducted which aimed to assess transcranial direct current stimulation (tDCS) efficacy on submaximal treadmill running time to exhaustion (TTE). TTE was not statistically different in the two groups.

Minor revisions:

1- Page 10, Section 2.3: State the statistical testing method which achieves 80% power.

We specified in the manuscript the statistical testing method. It now reads:

“The study sample size was estimated a priori using Student’s t-tests (see Supplementary Material) based on previous studies with amateur and competitive participants (cyclists and runners) [22,26,29–32].”

2- Section 2.7: The term "t-tests" is missing in the following sentence. “Comparisons between active and sham tDCS sessions were performed using paired Student’s [t-tests] and Wilcoxon tests.” Also specify the Wilcoxon tests: rank sum or signed rank.

Thank you. We corrected the sentence by adding the missing “t-tests”. We also specified the Wilcoxon tests used. This excerpt now reads:

“Baseline comparisons (age, weight, height, body fat, years of training, weekly training, VO2max, MAS, TTE on VO2max test, maximal heart rate, maximal respiratory exchange ratio) between groups (active/sham and sham/active) were performed using independent Student’s t-tests (normal distribution) and Wilcoxon rank-sum tests (non-normal distribution). Comparisons between active and sham tDCS sessions were then performed using paired Student’s t-tests and Wilcoxon signed-rank tests.”

3- To assist in the review process, add line numbering to the document.

We added line numbering to assist in the review process. 

Reviewer #2: As you mentioned the most problem in your study, is single session intervention so it seems that one session in not enough to have impact in physical performance, so it is better to emphasize in the conclusion to consider this issue in further study and try to have more sessions in their studies.

We thank the reviewer for the feedback and fully agree with issue raised about the single session aspect. To increase transparency about it in our manuscript we added the following in the limitations section;

“Our main limitations pertain to the tDCS protocol. We used a single session delivered before the TTE trial for feasibility constraints, while the concurrent application of tDCS with the targeted activity may have been more efficient [18,56]. This method also prevents the investigation of the cumulative effects of tDCS using several consecutive sessions.”

As well as the conclusion:

“To conclude, no beneficial effect of M1-tDCS on running performance has been identified. The potential effect of multiple sessions remains unknown and warrants further research.”

---

## [Decision Letter · Decision Letter 1]

1 Oct 2024

Can transcranial direct current stimulation (tDCS) over the motor cortex increase endurance running performance? A randomized crossover-controlled trial

PONE-D-24-16260R1

Dear Dr. Martens,

We’re pleased to inform you that your manuscript has been judged scientifically suitable for publication and will be formally accepted for publication once it meets all outstanding technical requirements.

Kind regards,

Abdolvahed Narmashiri

Academic Editor

PLOS ONE

Additional Editor Comments (optional):

Reviewers' comments:

Reviewer's Responses to Questions

**Comments to the Author**

1. If the authors have adequately addressed your comments raised in a previous round of review and you feel that this manuscript is now acceptable for publication, you may indicate that here to bypass the “Comments to the Author” section, enter your conflict of interest statement in the “Confidential to Editor” section, and submit your "Accept" recommendation.

Reviewer #1: All comments have been addressed

2. Is the manuscript technically sound, and do the data support the conclusions?

Reviewer #1: (No Response)

3. Has the statistical analysis been performed appropriately and rigorously? 

Reviewer #1: (No Response)

4. Have the authors made all data underlying the findings in their manuscript fully available?

Reviewer #1: (No Response)

5. Is the manuscript presented in an intelligible fashion and written in standard English?

Reviewer #1: (No Response)

6. Review Comments to the Author

Reviewer #1: All comments have been adequately addressed.

7. PLOS authors have the option to publish the peer review history of their article (what does this mean?). If published, this will include your full peer review and any attached files.

Reviewer #1: No

---

## [Editor Report · Acceptance letter]

23 Oct 2024

PONE-D-24-16260R1 

PLOS ONE

Dear Dr. Martens, 

I'm pleased to inform you that your manuscript has been deemed suitable for publication in PLOS ONE. Congratulations! Your manuscript is now being handed over to our production team.

Kind regards, 

on behalf of

Dr. Abdolvahed Narmashiri 

Academic Editor

PLOS ONE